

# Endophytic fungal community composition and function response to strawberry genotype and disease resistance

Hongjun Yang[1,2], Xu Zhang[1,2], Rui Wang[1], Quanzhi Wang[1,2], Yuanhua Wang[1,2], Geng Zhang[1,2], Pengpeng Sun[1,2], Bei Lu[1,2], Meiling Wu[1] and Zhiming Yan[1,2]

[1] College of Agronomy and Horticulture, Jiangsu Vocational College of Agriculture and Forestry, Zhenjiang, Jiangsu, China
[2] Jiangsu Engineering and Technology Center for Modern Horticulture, Zhenjiang, Jiangsu, China

## ABSTRACT

**Background**. Utilizing the plant endophytic microbiomes to enhance pathogen resistance in crop production is an emerging alternative method to chemical pesticides. However, research on the composition and role of microbial communities related to perennial fruit plants, such as the strawberry, is still limited.

**Methods**. We provide a comprehensive description of the composition and diversity of fungal communities in three niches (root, stem, and leaf) of three strawberry cultivars ('White Elves', 'Tokun', and 'Akihime') using internal transcribed spacer (ITS) rRNA amplicon sequencing and isolation culture methods. In addition, we also evaluated the disease tolerance ability of three strawberry cultivars to *Colletotrichum gloeosporioides* and *Alternaria alternata* through pathogenicity testing.

**Results**. 'White Elves' has stronger resistance to *Colletotrichum gloeosporioides*, and *Alternaria alternata*, followed by 'Tokun', while 'Akihime' has relatively weaker resistance to these pathogens. A total of 258 fungal strains were isolated from healthy strawberry plants and assigned to 34 fungal genera based on morphological and molecular characteristics analysis. Beneficial fungal genera such as *Trichoderma* and *Talaromyces* were more prevalent in 'White Elves', whereas common pathogenic fungi in strawberry, such as *Colletotrichum*, *Alternaria*, and *Fusarium*, were more prevalent in 'Akihime'. The composition and diversity of microbial communities vary among genotypes, and resistance to pathogens may play dominant roles in determining the microbial community structure. This study's results aid the biological control of strawberry fungal diseases and are useful for plant microbiome engineering in strawberry cultivation.

# INTRODUCTION

Endophytic microbiome are important parts of a plant's microbial community. Some endophytes provide benefits to their host plants by stimulating, $N_2$ fixation, phosphate solubilization, siderophores secretion, plant hormone synthesis, antifungal agent synthesis, and pathogen resistance (*Ashrafi et al., 2021*; *Bastías et al., 2018*; *Dang et al., 2024*; *Kandel,*

Corresponding authors
Xu Zhang, zhangxu@jsafc.edu.cn
Zhiming Yan, yanzhim@jsafc.edu.cn

*Joubert & Doty, 2017*; *Khan et al., 2021*; *Mei et al., 2021*; *Varga et al., 2020*). Endophytic microbes live in protected environments and can adapt to changes in nutrition, pH, and humidity; they are generally less affected by external environmental pressures than the host. They have a competitive advantage over rhizosphere and phyllosphere microbes (*Backman & Sikora, 2008*; *Harris, Bede & Tsuda, 2022*). However, the selection and assembly of an endophytic microbiota is a complex, dynamic, and continuous process, and its diversity and distribution are influenced by various factors, such as plant genotype, niche, geographical location, plant development stage, cultivation environment, and crop management methods (*Abdelrazek et al., 2020*; *Bonthond et al., 2022*; *Christian et al., 2016*; *Mina et al., 2020*; *Sangiorgio et al., 2022*; *Wang et al., 2024*; *Xiong et al., 2021*). Understanding the composition of endophytic microbial communities in plants is critical for the development and research of agricultural biofertilizers, biopesticides, and plants with improved traits.

Strawberry plants (*Fragaria × ananassa*) belong to the Rosaceae family and the *Fragaria* genus of perennial herbaceous plants. They are the most-cultivated small berries in the world and are a key global economic crop (*Denoyes et al., 2023*). The cultivation of strawberry is challenging because it is affected by numerous pathogens, such as root rot (*e.g.*, *Phytophthora* spp. and *Pythium* spp.), anthracnose (*Colletotrichum* spp.), wilt (*e.g.*, *Plectosphaerella* spp. and *Fusarium* spp.), and leaf spot disease (*Alternaria* spp. and *Botrytis* spp.) (*Ibañez et al., 2022*; *Ito et al., 2004*; *Marin & Peres, 2022*; *Poimala et al., 2021*; *Sharma et al., 2022*; *Tahat et al., 2022*; *Yang et al., 2023b*). The resistance of strawberry varieties to pathogens is controlled polygenically and varies among cultivars (*Lyzhin & Luk'yanchuk, 2024*). However, commercial strawberry cultivars are typically not resistant to most of these pathogens. Investigating the coevolution of endophytic microbial communities and host plants could provide a new approach for developing disease-resistant genotypes by enabling the manipulation of the relationship between plants and microbes (*Kaul et al., 2021*). In particular, biocontrol agents isolated from the host plant microbiome have superior efficacy compared with nonindigenous microbial inoculants (*Haney et al., 2015*; *Sangiorgio et al., 2022*).

Characterizing endophytic microbes in plants is a crucial step in achieving robust and sustainable plant microbiome engineering, successfully selecting beneficial microorganisms, and organic agriculture (*Nakielska et al., 2024*; *Purahong et al., 2018*). Therefore, in this study, we used internal transcribed spacer (ITS) rRNA amplicon sequencing and isolation culture methods to evaluate microbial differences in the root, stem, and leaf fungal communities for three strawberry genotypes ('White Elves', 'Tokun', and 'Akihime') with different disease tolerance. The aim of this work was to characterize the involvement of the plant genotype and it related disease tolerance on the fungal microbiomes of strawberry plants, with a focus on identifying pathogenic and beneficial microbes. Our research elucidates the impact of interaction between strawberry genotype and disease resistance on composition and diversity of endophytic fungal microbial communities, providing a theoretical basis for plant microbiome engineering in strawberry cultivation.

## MATERIALS & METHODS

### Sample collection

On November 17, 2021, healthy strawberry plant samples were collected from three cultivars (genotypes) in the strawberry greenhouse located in Jiangsu Agricultural Expo Park, Jiangsu Province, China (32°02′N, 119°26′E). The greenhouse conditions were as follows: daytime temperature 23–25 °C, nighttime temperature 8–10 °C, humidity: 60%–80%, row spacing: 30 cm, and plant spacing 20 cm. The cultivars included 'White Elves', 'Tokun', and 'Akihime', with 10 plants from each cultivar were randomly selected for collection.

The whole strawberry plant was excavated, and the roots, stems, and leaves (three niches) of each plant were placed separately in sterile plastic bags and transported to the laboratory on ice for immediate processing. A total of 90 samples were analyzed: three niches (root, stem, and leaf) for ten plants of the three cultivars ('White Elves', 'Tokun', and 'Akihime').

### Assessing the disease tolerance ability of three strawberry cultivars

Strains of *Colletotrichum gloeosporioides* and *Alternaria alternata*, previously isolated from diseased strawberry plants, were cultured on potato dextrose agar (PDA) plates in the laboratory for 5 d at 25 °C in the dark. Ten healthy strawberry leaves from each of the three cultivars were selected, washed with tap water, soaked in 70% (v/v) ethanol for 30 s, and then washed three times with sterile water. The leaves were lightly pricked on both sides using a sterilized inoculation needle, and 0.5 cm diameter culture blocks containing 5 d-old pathogen mycelia were placed on the wounds. The inoculation sites were wrapped with absorbent paper soaked in water. The control group was treated with blank PDA medium instead of mycelium blocks. The inoculation sites were wrapped with absorbent paper soaked with water. The inoculated leaves were placed in a tray, covered with two layers of damp sterile gauze, sealed with cling film, and kept at 25 °C in the dark. After 5 d, the incidence of disease on the strawberry leaves was checked. The diameter of the lesions was measured using the cross method, and the average lesion size was calculated.

### Sample preparation, fungal isolation and DNA extraction

In the laboratory, each sample was subjected to strict surface disinfection per our previous established protocol (*Yang et al., 2023a*). Briefly, first the surface of each strawberry plant was rinsed with tap water to remove soil residue and dust. Each sample was cut into small pieces and mixed thoroughly, and 2 g of each sample was retained. Epiphytic microbes were removed by mixing the sample with 100 mL of sterile water and two drops of polysorbates 20 (Tween 20) (Macklin, Shanghai, China); the mixture was shaken at 220 rpm for 20 min at 25 °C. It was then treated with sterile water for 20 s, with 70% (v/v) ethanol for 30 s, and 2.5% (v/v) sodium hypochlorite solution for 2 min. Finally, the sample was rinsed three to times with sterile water and dried. The root, stem, and leaf samples were cut into 0.25 cm fragments and divided into two parts for fungal isolation and DNA extraction.

To isolate fungi, five random fragments from each sample were taken and placed them on potato dextrose agar (PDA) plates containing ampicillin (50 mg/L) and rifampicin (50 mg/L). A total of 270 plates (3 cultivars × 3 niches × 10 plants × 3 replicates) were cultured

at 25 °C for three replicates. After 3–5 d, when mycelium appeared from the tissue block, a small piece of the edge of the medium was carefully removed and transferred together with the mycelium to a new plate containing PDA plate. These pure fungal cultures were preliminarily classified into morphological taxa on the basis of the type of mycelium, colony color, and growth rate. The mycelia were allowed to grow for 7 d after transferral, and the mycelium DNA was then extracted using the DNAsecure Plant Kit (Tiangen, Beijing, China) according to the manufacturer's instructions.The internal transcribed spacers (ITS1/ITS4) were then amplified following our previously outlined protocols to identify the isolated species of endophytic fungus (*Yang et al., 2023a*). Additionally, the *beta-tubulin* (*BenA*) gene was amplified to further confirm the species identification within the genus *Talaromyces* (*Zhang et al., 2021*), and the *tef1* and *RPB2* genes were amplified for species confirmation within the genus *Trichoderma* (*Druzhinina & Kubicek, 2005*). The polymerase chain reaction (PCR) products were sequenced, and the resulting sequences were compared with those in the NCBI database for species identification.

An additional two g of short fragments were collected from each sample for DNA extraction using the aforementioned reagent kit according to the manufacturer's instructions and stored at −20 °C prior for further analysis. The DNA concentration and purity were quantified using a NanoDrop 2000 spectrophotometer (Thermo Fisher Scientific, Waltham, MA, USA).

## PCR amplification and sequencing

The fungal ITS1 regions of the nuclear ribosomal ITS rRNA gene were amplified using primers ITS1F (5′-CTTGGTCATTTAGAGGAAGTAA-3′) and ITS2 (5′-GCTGCGTTCTTCATCGATGC-3′) (*Maarastawi et al., 2018*; *Santos-Medellin et al., 2017*; *Xiong et al., 2017*). PCR was carried out using five µL of 5× buffer, two µL of 2.5 mM deoxynucleotide triphosphates (dNTPs), one µL of each primer (10 µM), 0.25 µL of Fast pfu DNA Polymerase (Q5® High-Fidelity DNA Polymerase (NEB)) (5U/µL), and one µL of template DNA. The final volume was adjusted to 25 µL with ddH$_2$O. The PCR program included an initial denaturing at 98 °C for 5 min, followed by 25 cycles at 98 °C for 30 s, 53 °C for 30 s, and 72 °C for 45 s with a final extensionstep at 72 °C for 5 min and ending at 4 °C. The PCR products were analyzed by 2% agarose gel electrophoresis. Amplicons were subjected to paired-end sequencing on an Illumina NovaSeq sequencing platform with PE250 (Shanghai Personal Biotechnology, Shanghai, China). Raw sequence reads were deposited at the Sequence Read Archive of the National Center for Biotechnology Information (accession ID: PRJNA1116726).

## Bioinformatic analysis

Data were collected as previously described in *Yang et al. (2023a)*. Specifically, after demultiplexing, we used FLASH v.1.2.11 to merge the sequences (*Magoc & Salzberg, 2011*) and performed quality filtering using fastp v.0.19.6 software (*Chen et al., 2018*). High-quality sequences were denoised using the DADA2 (*Callahan et al., 2016*) or Deblur (*Amir et al., 2017*) plugins in Qiime2 v. 2020.0 (*Bolyen et al., 2019*) to obtain the amplicon sequence variants (ASVs). Taxonomy assignments for the fungal ASVs were performed

using the classify-sklearn naive Bayes taxonomy classifier in the feature-classifier plugin (*Bokulich et al., 2018*) with reference to the UNITE database v.8.0 (*Koljalg, Nilsson & Abarenkov, 2013*).

ASVs classified as "norank" or "unclassified_k__Fungi" were excluded from the fungal ASV table according to our previous protocols (*Yang et al., 2023a*). Alpha diversity indices,such as Chao1 and Shannon diversity indices, were calculated from the ASV table using Mothur v.1.30.2. Principal coordinate analysis (PCoA) based on Bray–Curtis distances was performed in QIIME and visualized using R v.3.3.1. Potential functions of the fungal communities among different cultivars in different strawberry niches were estimated (FUNGuild, https://www.funguild.org/) (*Nguyen et al., 2016*; *Toju et al., 2016*) calculations. ASVs assigned to a guild with a confidence ranking of "highly probable" or "probable" were retained in the analysis, whereas those ranked as "possible" were considered unclassified following previously published protocols (*Yang et al., 2023a*).

### Statistical analysis

The relative abundances of fungi at phylum and genus levels, alpha diversity indices, and the relative abundance of fungi among genotypes (cultivars) for each plant compartments (niches), as well as the disease spot diameter of three strawberry cultivars inoculated with pathogens, were compared using one-way analysis of variance (ANOVA) followed by Tukey's honestly significant difference (HSD) test ($P < 0.05$). Additionally, the relative abundances of the most abundant fungal taxa in each plant compartment were also compared using one-way ANOVA followed by Tukey's HSD test ($P < 0.05$). All statistical analyses were performed using SPSS ver. 20.0 (IBM, Armonk, NY, USA).

## RESULTS

### Tolerance of three strawberry genotypes to *C. gloeosporioides* and *A. alternata*

When inoculated with *C. gloeosporioides* and *A. alternata*, 'White Elves' exhibited the lowest disease incidence of 20% and 70%, respectively. In contrast, the disease incidence of 'Akihime' inoculated with *C. gloeosporioides* and *A. alternata* was the highest, which was 50% and 90%, respectively. The disease incidence of *C. gloeosporioides* inoculation in 'Tokun' was only 10%, while the disease incidence of *A. alternata* inoculation was 100% (Fig. S1). We further measured the disease spot diameter and found that, following inoculation with both pathogens, 'White Elves' had the smallest disease spot diameter, whereas 'Akihime' had the largest disease spot diameter. In particular, after infection with *C. gloeosporioides*, the disease spot diameter on the 'White Elves' and 'Tokun' leaves was significantly smaller than that of 'Akihime' (Figs. 1A, 1B). Together, these results indicated that the 'White Elves' has stronger resistance to *C. gloeosporioides* and *A. alternata*, while 'Akihime' has relatively weaker resistance to these pathogens.

### Fungal microbiome structure and diversity were associated with host genotype and disease resistance

To examine the associations of host genotype and disease resistance on the strawberry fungal microbiome, we analyzed the fungal communities in three niches (root, stem, and

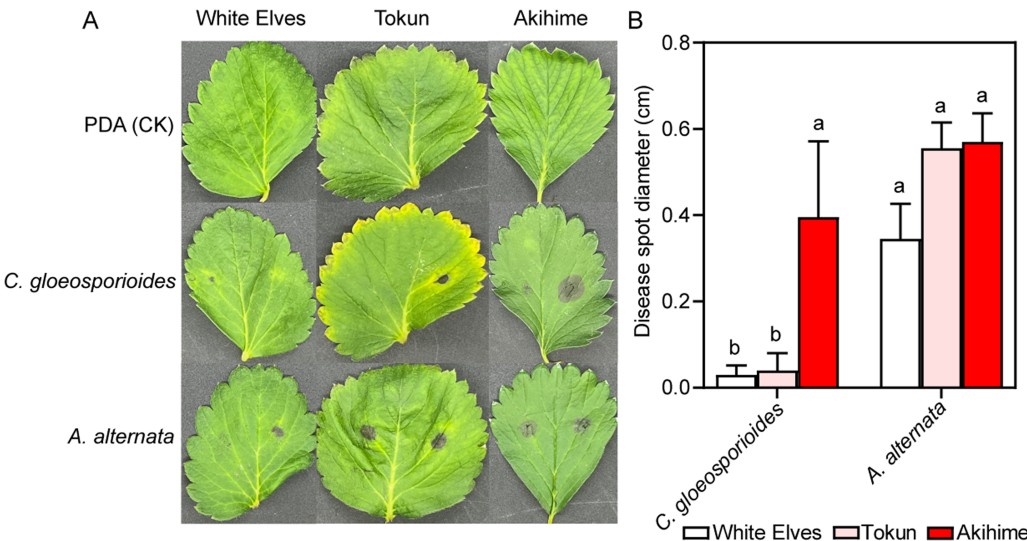

**Figure 1 Disease symptoms and disease spot diameter.** (A) Disease symptoms on leaves of three straw-berry cultivars ('White Elves', 'Tokun', and 'Akihime') inoculation with *C. gloeosporioides* and *A. alternata*. (B) Box plots with error bars of the disease spot diameter (cm) measured on 10 leaves per cultivar. Different letters above the bars indicate statistically significant differences according to Tukey's HSD test ($P <$ 0.05, one-way ANOVA).

leaf) of three strawberry cultivars ('White Elves', 'Tokun', and 'Akihime') with different disease resistance by sequencing the ITS1 region. Through quality filtering, denoising, merging, and removing chimeras, we obtained a total of 5, 541, 374 sequences, which could be assembled into 7, 311 fungal ASVs.

The Shannon alpha diversity index and Chao1 community richness index suggested that fungal community diversity of each niche (root, stem, and leaf) did not differ between cultivars, but the 'Tokun' richness was lower than the two other cultivars (Figs. 2A–2B). In the roots, the three cultivars exhibited similar diversity and richness (Figs. 2A–2B). In the stem the leaves, the three cultivars showed similar diversity; while the richness of the 'Tokun' was significantly lower than those of both 'White Elves' and 'Akihime' (Figs. 2A–2B).

PCoA combined with an analysis of similarities (ANOSIM) indicated that genotype had a stronger association than niche in fungal communities (genotype $R = 0.309$, $P < 0.001$; niche $R = 0.186$, $P < 0.001$) (Fig. 3A). The fungal community structure differed the most in the stem ($R = 0.601$, $P < 0.001$), followed by the leaf ($R = 0.549$, $P < 0.001$) and then the root ($R = 0.353$, $P < 0.001$) (Fig. S2). A total of three common core ASVs in the fungal community were identified, and the number of unique fungal taxa from root to stem to leaf increased sequentially among the three cultivars. These three core taxa were *Rhodotorula*, unclassified Ascomycota, and *Plectosphaerella* (Fig. 3B).

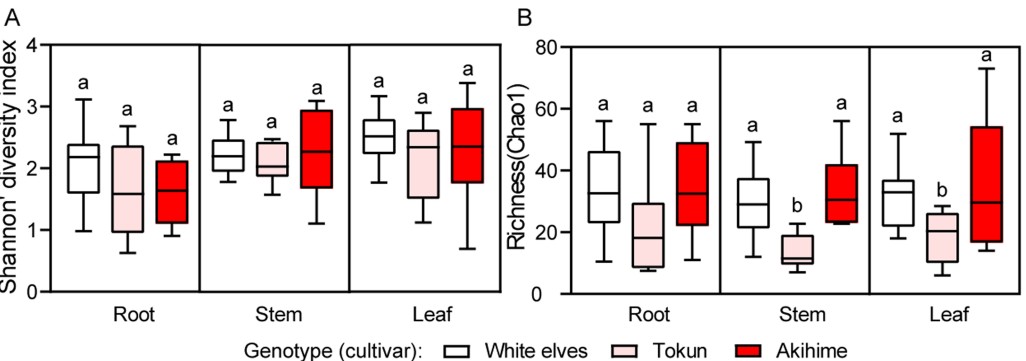

**Figure 2** **Alpha diversity and richness of fungal communities among three strawberry cultivars in roots, stems, and leaves ($n = 10$).** (A) Diversity (Shannon index) of the fungal community. (B) Richness (Chao1) of the fungal community. Different letters above the bars indicate statistically significant differences according to Tukey's HSD test ($P < 0.05$, one-way ANOVA).

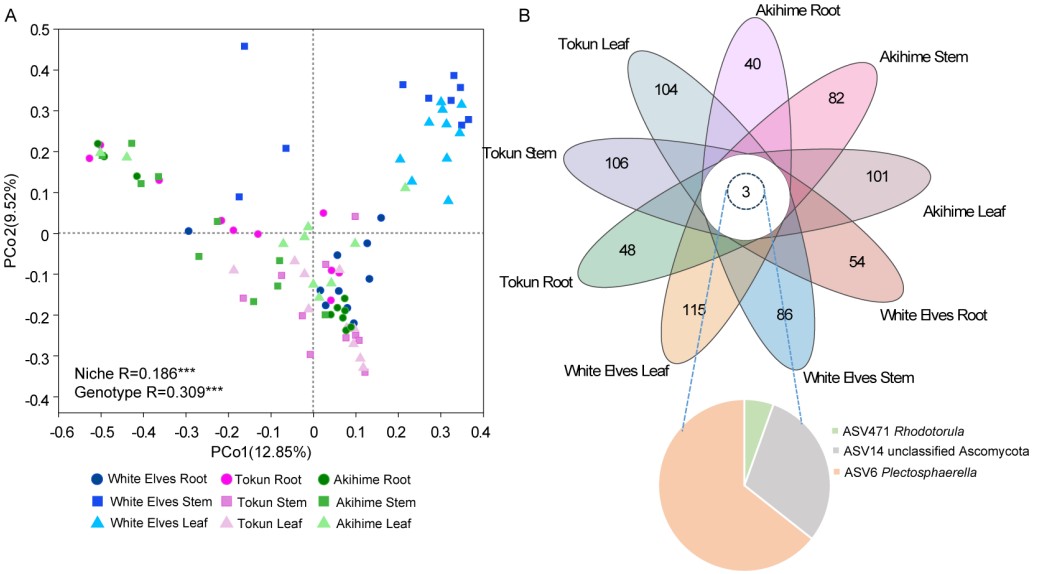

**Figure 3** **Endophytic fungal community exhibit a distinct host genotype response.** (A) PCoA for fungal communities of three strawberry cultivars with Bray–Curtis dissimilarities ($n = 10$). ANOSIM conducted to test for differences in community composition resulting from niche and cultivars. $R$ values presented and labeled with asterisks: ***$P < 0.001$. (B) The flower diagram showing the number of fungal ASVs shared among the three strawberry cultivars ($n = 10$).

## Beneficial fungal taxa were more abundant in 'White Elves' than in 'Akihime'

All seven dominant ($\geq 0.1\%$ relative abundance) fungal phyla were observed in this study (Fig. 4A). In the roots, Basidiomycota was more abundant in 'Tokun' than in 'Akihime' ($P < 0.05$, Tukey's HSD). In the stems, Ascomycota was more abundant in 'White Elves' and 'Akihime' than in 'Tokun', whereas Basidiomycota was more abundant in 'Tokun'

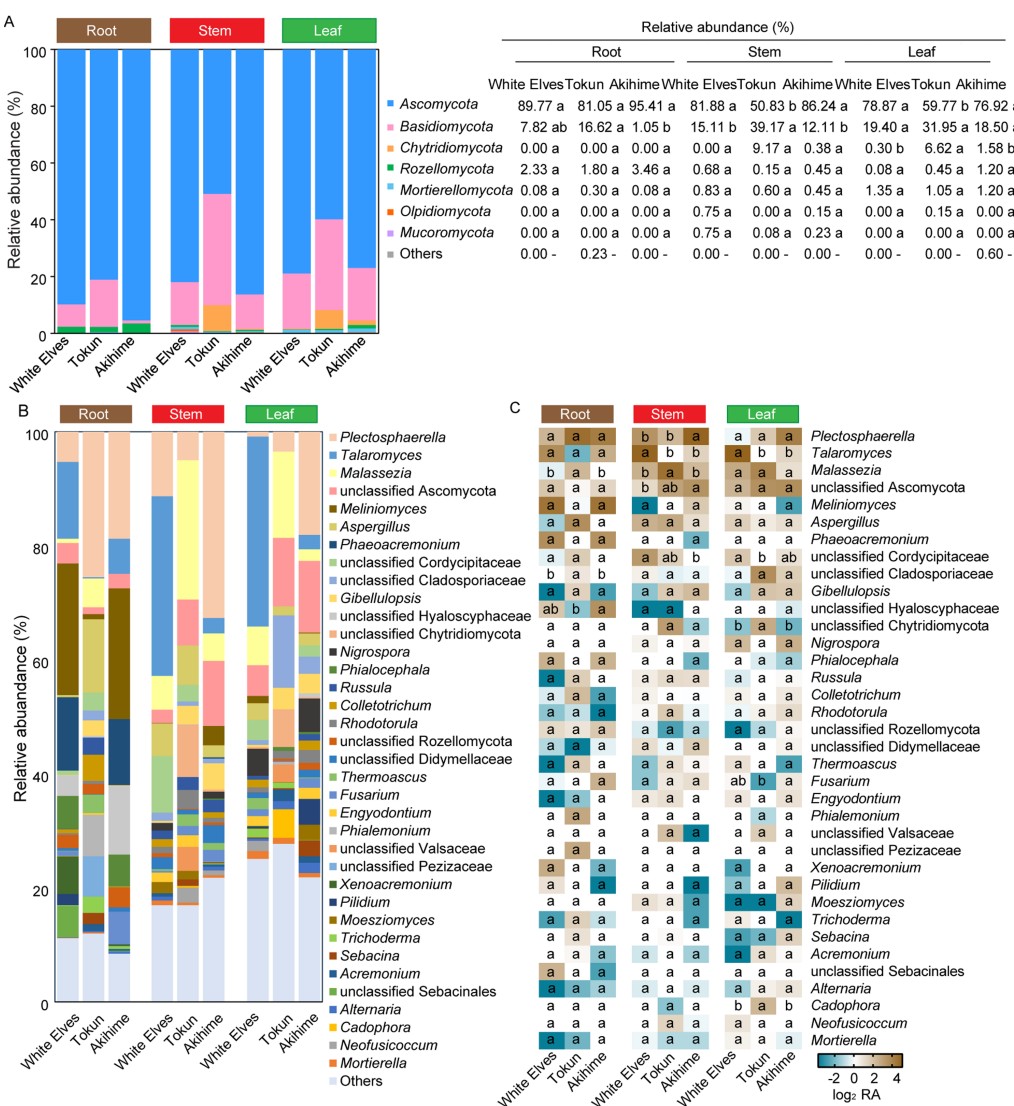

**Figure 4   Relative abundances of fungal community compositions.** (A) Relative abundance of most abundant ($> 0.1\%$) fungal phyla among three strawberry cultivars in roots, stems, and leaves ($n = 10$). (B, C) Relative abundance of dominant fungal genera among three strawberry cultivars in roots, stems, and leaves ($n = 10$). Figure depicts fungal genera with relative abundance of $> 0.5\%$. Cell colors represent the $\log_2$ fold change in relative abundance compared with control treatment; brown and cyan indicate increasing and decreasing trends, respectively. Different letters indicate statistically significant differences according to Tukey's HSD test ($P < 0.05$, one-way ANOVA).

than in 'White Elves' or 'Akihime' ($P < 0.01$, Tukey's HSD). In the leaves, Ascomycota was more abundant in 'White Elves' than in 'Tokun' ($P < 0.05$, Tukey's HSD), whereas Chytridiomycota was more abundant in 'Tokun' than in all other cultivars ($P < 0.01$, Tukey's HSD) (Fig. 4A).

A total of 36 dominant fungal genera with a relative abundance of $\geq 0.5\%$ were identified (Figs. 4B–4C). In the roots, *Malassezia* and unclassified Cladosporiaceae were more abundant in 'Tokun' than in the other cultivars ($P < 0.01$, Tukey's HSD). In the
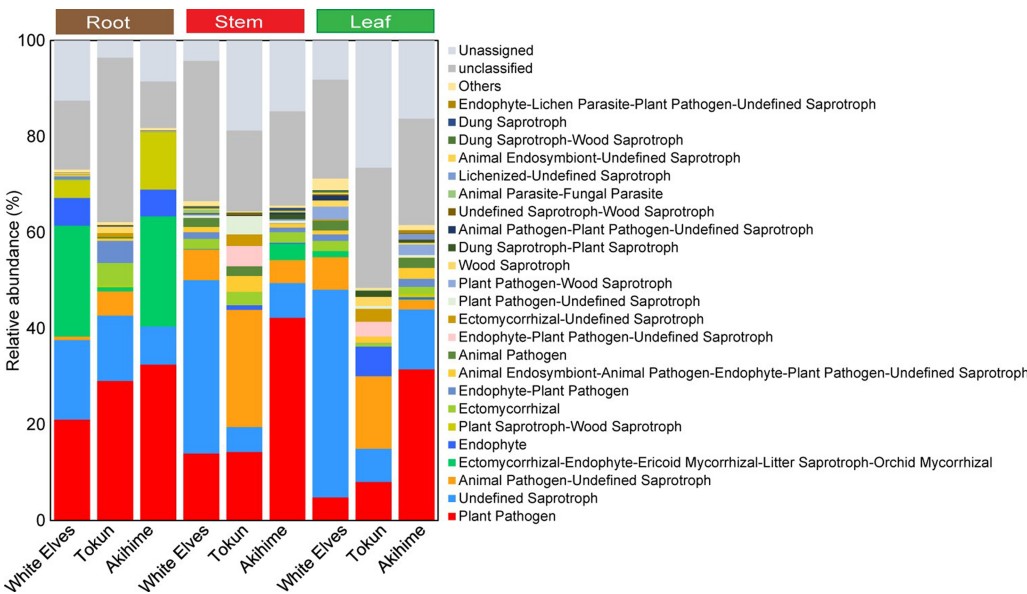

**Figure 5** Relative abundance of fungal predicted functional groups (guilds) among three different cultivars in the three niches of strawberry roots, stems, and leaves inferred using FUNGuild ($n = 10$).

stems, *Plectosphaerella* had the greatest abundance in 'Akihime', *Talaromyces* had the greatest abundance in 'White Elves', and *Malassezia* had the greatest abundance in 'Tokun' ($P < 0.01$, Tukey's HSD). In the leaves, *Talaromyces* had the greatest abundance in 'White Elves', whereas *Fusarium* was more abundant in 'Akihime' than in 'Tokun' ($P < 0.01$, Tukey's HSD) (Figs. 4B–4C).

## Fungal functional guilds of the three strawberry genotypes with different resistance

ASVs assigned to a guild with a confidence ranking of "highly probable" or "probable" were retained in the analysis, whereas those with a confidence ranking of "possible" were regarded as unclassified (*Cregger et al., 2018*). Twenty-four fungal functional groups colonized the different niches of the strawberry plants (Fig. 5, Table S1). In the roots, the relative abundance of animal pathogen-undefined saprotroph in 'Tokun' was higher than those in other cultivars ($P < 0.01$, Tukey's HSD). In the stems, 'Akihime' had the highest relative abundance of plant pathogen, 'White Elves' had the highest relative abundance of undefined saprotroph, and 'Tokun' had the highest relative abundance of animal pathogen-undefined saprotroph relative to the other cultivars ($P < 0.01$, Tukey's HSD). In the leaves, 'Akihime' had the highest relative abundance of plant pathogen, 'White Elves' had the highest relative abundance of undefined saprotroph, and 'Tokun' had the highest relative abundance of endophyte relative to the other cultivars ($P < 0.01$, Tukey's HSD) (Fig. 5). Notably, the relative abundance of plant pathogens in 'Akihime' was significantly higher, suggesting that 'Akihime' may be more susceptible to strawberry plant diseases.

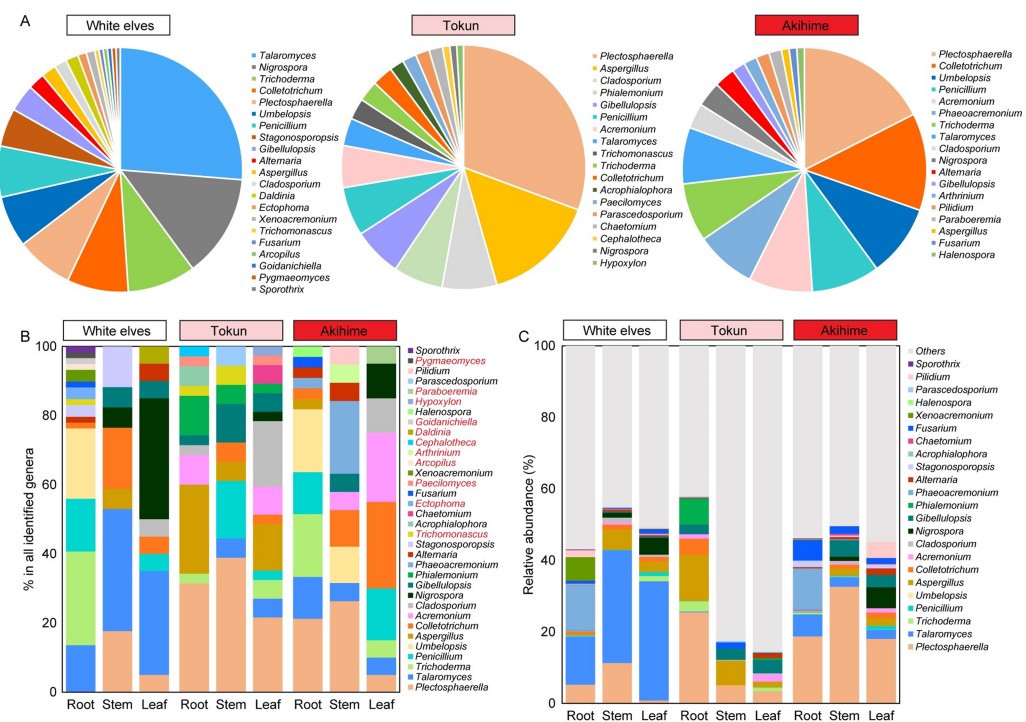

**Figure 6** **Comparison of the results of isolation culture and high-throughput sequencing identification of endophytic fungal communities.** (A) Proportion of overall community composition of fungal endophytes isolated in different strawberry cultivars. (B) Relative abundance of fungal genera isolated from roots, stems, and leaves of three strawberry cultivars. (C) Relative abundance of corresponding fungal genera in ITS high-throughput sequencing.

## Comparison of fungal isolation taxa and high-throughput sequencing results among the three strawberry genotypes with different resistance

Results on the fungal community compositions of the three strawberry cultivars differed depending on whether they were obtained using isolation culture or high-throughput sequencing. In the isolation culture, a total of 258 fungal strains were isolated from three strawberry cultivars, and 34 genera were identified. Among them, 21, 18, and 18 genera were identified in 'White Elves', 'Tokun', and 'Akihime', respectively (Fig. 6A). The 23 genera identified by isolation culture were also detected in high-throughput sequencing methods, whereas 11 genera were only detected in the isolation and cultivation methods, namely *Trichomonascus*, *Ectophoma*, *Paecilomyces*, *Arcopilus*, *Arthrinium*, *Cephalotheca*, *Daldinia*, *Goidanichiella*, *Hypoxylon*, *Paraboeremia*, and *Pygmaeomyces* (Figs. 6B, 6C).

In the fungal isolation and cultivation results, the strains *Plectosphaerella*, *Talaromyces*, and *Trichoderma* had the highest abundance, accounting for 18.57%, 12.47%, and 6.51% of the samples, respectively. Correspondingly, in high-throughput sequencing analysis, *Plectosphaerella*, *Talaromyces*, and *Trichoderma* accounted for 13.41%, 9.99%, and 0.69% respectively (Fig. 6B, 6C; Tables S2, S3).

For 'White Elves', the genera *Talaromyces* had the highest relative abundance and *Plectosphaerella* had the highest relative abundance in 'Tokun' and 'Akihime'; this was consistent with the results of high-throughput ITS sequencing (Figs. 6A–6C; Table S2). According to our results, beneficial fungal genera such as *Trichoderma* and *Talaromyces* were more prevalent in 'White Elves', whereas common pathogenic fungi in strawberry, such as *Colletotrichum*, *Fusarium*, and *Alternaria*, were more prevalent in 'Akihime'; this is consistent with cultivar disease resistance, fungal functional predictions, and field observations (Fig. 6B, Table S2). The genus *Talaromyces* in the 'White Elves' mainly comprised *Talaromyces kabodanensis* and *Talaromyces wushanicus*, the genus *Trichoderma* in the 'White Elves' mainly comprised *Trichoderma hamatum* and *Trichoderma asperellum* (Table S4).

## DISCUSSION

### The effect of plant genotype with different resistance on the endophytic fungal microbiomes of strawberry

Endophytic fungi can colonize healthy plant tissues and organs, serving as significant sources of new microbial resources and natural bioactive products. They play a crucial role in the prevention and control of plant diseases (*Hu et al., 2020*; *Mina et al., 2020*; *Sangiorgio et al., 2022*; *Wang et al., 2022*). Consistent with previous studies, this study highlights the importance of strawberry niches and genotypes in shaping the taxonomic and functional composition of endophytic fungal communities (*Sangiorgio et al., 2022*; *Wang et al., 2024*).

In the present study, we found that the diversity of endophytic fungi in strawberry roots, stems, and leaves was relatively consistent across the cultivars. However, the fungal community richness in 'Tokun' was lower than that of the other cultivars, which may which may reflect differences in the tolerance or ecological adaptability of the cultivars to pathogens. Fungal taxonomic composition varies greatly among the strawberry cultivars with different resistance. At the genus level, in roots and stems, *Malassezia* is the most abundant in 'Tokun' than in other cultivars. In the stems, *Plectosphaerella* is the most abundant in 'Akihime'. In the stems and leaves, *Talaromyces* is more abundant in 'White Elves' than in 'Tokun' and 'Akihime'. In the leaves, *Fusarium* is more abundant in 'Akihime' than in 'Tokun'. The specific role of *Malassezia* in strawberry plants is unknown. *Malassezia* typically exists in various ecosystems, such as human skin, human gut, and marine environments, and has mutualistic or pathogenic interactions with its host (*Ianiri, LeibundGut-Landmann & Dawson, 2022*). Some species of *Plectosphaerella* and *Fusarium* are strawberry pathogens that cause strawberry wilt (*Tahat et al., 2022*; *Yang et al., 2023b*), although no evident disease symptoms were observed in our samples. Other studies have identified potential fungal pathogens species, such as *Plectosphaerella cucumerina*, *Botrytis caroliniana*, *Fusarium incarnatum*, *Alternaria tenuissima*, and *Alternaria alternata*. in the endophytic fungal community of strawberry (*Sangiorgio et al., 2022*; *Wang et al., 2024*). *Talaromyces* species are worldwide filamentous fungi with the ability to exert antagonistic activities against plant pathogens and are used for biological control of fungal phytopathogens, such as *Sclerotium rolfsi*, *Botrytis cinerea*, *Fusarium solani*, *F. oxysporum*,

*Rhizoctonia solani*, and *Macropomina phaseolina* (*Abdel-Rahim & Abo-Elyousr, 2018*; *Coulibaly et al., 2022*; *Farhat et al., 2023*).

At the species level, *Plectosphaerella cucumerina* was the most isolated endophytic fungus, followed by *Trichoderma hamatum*, *Aspergillus fumigatus*, *Umbelopsis vinacea*, *Colletotrichum siamense*, and *Talaromyces kabodanensis*. *Plectosphaerella cucumerina*, the core fungal microbiome, has a significantly lower abundance in 'White Elves' compared to 'Tokun' and 'Akihime'. Research has reported that *Plectosphaerella cucumerina* can cause strawberry crown and root rot and wilt disease (*Hassan & Chang, 2021*; *Yang et al., 2023b*). In addition, it was also found in the endophytic fungal community of strawberry core microbiome, although no disease symptoms were observed. (*Sangiorgio et al., 2022*). However, the relative abundance of *Trichoderma hamatum*, *Umbelopsis vinacea*, and *Talaromyces kabodanensis* in the 'White Elves' is significantly higher than that of 'Tokun' and 'Akihime'. *Trichoderma hamatum* is an effective biological control agent in strawberry potting systems, which can suppress strawberry root rot and increase strawberry dry weight (*Leandro, Ferguson & Louws, 2007*). The relative abundance of *Colletotrichum siamense* in the 'Akihime' is significantly higher than that of 'White Elves' and 'Tokun'. Research has shown that *Colletotrichum siamense* can cause strawberry anthracnose crown rot and anthracnose disease (*Jian et al., 2021*; *Westrick & Salvas, 2024*).

This suggests that the differences in endophytic fungal communities among different genotypes of strawberry may be related to their disease tolerance. Cultivars with strong disease tolerance have lower pathogen abundance and higher abundance of taxa that inhibit pathogens, which may be an important contribution of the microbiota unique to tolerant genotypes to the defense of strawberry plants against pathogens (*Sangiorgio et al., 2022*). Fungal pathogens in cultivars with poor disease resistance may invade host tissues and outcompete other fungal species, leading to a decrease in fungal microbial species in the cultivar. Perhaps just as the presence of fungal pathogens may alter the composition and diversity of microbial communities inhabiting hybrid poplar trees (*Cregger et al., 2018*).

## Fungal functional guilds of the three strawberry genotypes

The FUNGuild function prediction results showed that 24 fungal functional groups are colonized in different niches of strawberry plants. In the roots, the relative abundance of animal pathogen-undefined saprotroph in 'Tokun' was higher than that in other cultivars. In the stems and leaves, 'Akihime' had the highest relative abundance of plant pathogens, whereas 'White Elves' have the highest relative abundance of undefined saprotroph compared with other cultivars. The plant pathogens that were dominant in plant tissues, albeit primarily the stems and roots, significantly differed between different cultivars, with 'Akihime' having the highest relative abundance. This may be related to the disease resistance of various strawberry cultivars (*Cregger et al., 2018*). Although our strawberry plants exhibited no symptoms of disease, under appropriate conditions, some endophytes may cause disease symptoms in plants after a long incubation period, causing plant disease. Future study should focus on the impact of pathogen to strawberry microbiome diversity and structure to better understand their role in plant health and disease dynamics.

 

### Fungal isolation and high-throughput sequencing analysis

After tissue isolation, a total of 258 fungal strains were isolated from three strawberry cultivars, and 34 genera were identified. Among these, 23 genera were only detected by the high-throughput sequencing methods, whereas 11 genera were only detected by the isolation and cultivation methods. Plant tissue isolation and high-throughput sequencing were used in conjunction in our investigation of endophytes in plants, which allowed us gain a more comprehensive understanding of plant endophytic microbes and facilitate subsequent research on their functions and characteristics (*Impullitti & Malvick, 2013*). For 'White Elves', the genera *Talaromyces* had the highest relative abundance, whereas *Plectosphaerella* had the highest relative abundance in 'Tokun' and 'Akihime'; this was consistent with the results of high-throughput ITS sequencing. According to our research results, beneficial fungal genera such as *Trichoderma* and *Talaromyces* had a higher proportion in 'White Elves', whereas common pathogenic fungi in strawberry, such as *Colletotrichum*, *Fusarium*, and *Alternaria*, was more abundant in 'Akihime', which is consistent with fungal functional predictions and field observations.

Some species of *Plectosphaerella* and *Fusarium* are pathogens that cause strawberry wilt (*Tahat et al., 2022*; *Yang et al., 2023b*), some species of *Colletotrichum* are pathogens that cause strawberry anthracnose (*Jian et al., 2021*; *Soares et al., 2021*), and some species of *Alternaria* are pathogens that cause strawberry fruit rot and *Alternaria* black spot (*Ito et al., 2004*; *Li et al., 2023*). These results indicate differences in the potential pathogenic fungi among different strawberry cultivars, with 'Akihime' having the most latent pathogenic fungi; this is consistent with the observed poor disease resistance in 'Akihime' cultivars during production.

previous studies also found that 'Akihime' has a higher relative abundance of potential pathogenic fungi, such as *Fusarium incarnatum* and *Rhizoctonia fragariae, etc.*) compared wild strawberry cultivar *F. nilgerrens* is (*Wang et al., 2024*). The *Trichoderma* species in the 'White Elves' were mainly *T. hamatum*, *T. asperellum*, and *T. viride*. *T. asperellum* inhibits mycelial growth, especially the conidia germination of *Botrytis cinerea*. Due to its volatile organic compounds damaging the cell membrane permeability and integrity of *B. cinerea*, it has been used as a biological control agent (*Fan et al., 2024*; *Sanz et al., 2004*). *T. viride* is highly effective in protecting strawberry from the invasion of the black root rot pathogen *Rhizoctonia fragariae* and can improve plant growth, on several parameters, and fruit quantity (*Abied et al., 2023*). *T. hamatum* and their secondary metabolites have antimicrobial activity against the pathogens of plant root rot (*Liu et al., 2023*).

## CONCLUSIONS

This work highlights the effects of genotype and niche on the composition and diversity of strawberry fungal communities; significant differences in fungal taxonomic composition were observed among different strawberry cultivars, which may be related to their resistance to pathogens. 'White Elves' has stronger resistance to *C. gloeosporioides* and *A. alternata*, while 'Akihime' has relatively weaker resistance to these pathogens. In this study, beneficial fungal genera such as *Trichoderma* and *Talaromyces* were more prevalent in 'White Elves',

whereas common pathogenic fungi in strawberry, such as *Colletotrichum*, *Fusarium*, and *Alternaria*, were more prevalent in 'Akihime', which is consistent with fungal functional predictions and field observations. The results of this study can provide a theoretical basis and resources for the biological control of strawberry fungal diseases and the exploration and manipulation of the relationship between plants and microbes to improve plant disease resistance.

### Funding
This work was supported by the Natural Science Foundation of the Jiangsu Higher Education Institutions of China (23KJB210009), the Science Fund of Jiangsu Vocational College of Agriculture and Forestry (2021kj22), the Yafu Technology Innovation Team (2024kj02) and the Unveiling and Leading Projects (2022kj05; 2023kj14) of Jiangsu Vocational College of Agriculture and Forestry. The funders had no role in study design, data collection and analysis, decision to publish, or preparation of the manuscript.

### Grant Disclosures
The following grant information was disclosed by the authors:
Natural Science Foundation of the Jiangsu Higher Education Institutions of China: 23KJB210009.
Science Fund of Jiangsu Vocational College of Agriculture and Forestry: 2021kj22.
Yafu Technology Innovation Team: 2024kj02.
Unveiling and Leading Projects: 2022kj05, 2023kj14.

### Competing Interests
The authors declare there are no competing interests.

### Author Contributions
- Hongjun Yang conceived and designed the experiments, performed the experiments, analyzed the data, prepared figures and/or tables, authored or reviewed drafts of the article, and approved the final draft.
- Xu Zhang conceived and designed the experiments, performed the experiments, analyzed the data, prepared figures and/or tables, authored or reviewed drafts of the article, and approved the final draft.
- Rui Wang performed the experiments, authored or reviewed drafts of the article, and approved the final draft.
- Quanzhi Wang conceived and designed the experiments, authored or reviewed drafts of the article, and approved the final draft.
- Yuanhua Wang conceived and designed the experiments, authored or reviewed drafts of the article, and approved the final draft.
- Geng Zhang conceived and designed the experiments, authored or reviewed drafts of the article, and approved the final draft.

- Pengpeng Sun analyzed the data, authored or reviewed drafts of the article, and approved the final draft.
- Bei Lu analyzed the data, authored or reviewed drafts of the article, and approved the final draft.
- Meiling Wu performed the experiments, authored or reviewed drafts of the article, and approved the final draft.
- Zhiming Yan conceived and designed the experiments, authored or reviewed drafts of the article, and approved the final draft.

### DNA Deposition

The following information was supplied regarding the deposition of DNA sequences:

The raw sequence reads are available at the Sequence Read Archive of the National Center for Biotechnology Information: PRJNA1116726.

### Data Availability

Raw data for disease spot diameter (Fig. 1B), relative abundance of fungal species isolated from roots, stems, and leaves of three strawberry cultivars, ITS sequences, and the number of fungal strains isolated from roots, stems, and leaves of three strawberry cultivars are available in the Supplementary File.

### Supplemental Information

Supplemental information for this article can be found online at http://dx.doi.org/10.7717/peerj.19383#supplemental-information.

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
