# Peer review of "Endophytic fungal community composition and function response to strawberry genotype and disease resistance"

_PeerJ, doi:10.7717/peerj.19383_

## Round 0.1 · original submission · Major Revisions

Your manuscript needs major revisions to address the concerns of the reviewers. Pay attention to reducing the high similarity rate. You should also explain your method in more detail.

·

Basic reporting

The introduction section of the manuscript outlines the significance of studying endophytic fungal communities in relation to strawberry genotypes and their disease resistance, highlighting the potential impact on agricultural practices.
The data, obtained by authors are valuable, but some recent studies were missing. These two papers could extend the Intro section:
Nakielska M, Feledyn-Szewczyk B, Berbeć AK, Frąc M. Microbial Biopreparations and Their Impact on Organic Strawberry (Fragaria x ananassa Duch.) Yields and Fungal Infestation. Sustainability. 2024; 16(17):7559. https://doi.org/10.3390/su16177559
Wang Z, Dai Q, Su D, Zhang Z, Tian Y, Tong J, Chen S, Yan C, Yang J and Cui X (2024) Comparative analysis of the microbiomes of strawberry wild species Fragaria nilgerrensis and cultivated variety Akihime using amplicon-based next-generation sequencing. Front. Microbiol. 15:1377782. doi: https://doi.org/10.3389/fmicb.2024.1377782

NCBI Bioproject is valid, but SRA and Biosample records are empty (shows “No items found”). Maybe there is some problem with public access with them? See the following URLs:
https://ncbi.nlm.nih.gov/sra?linkname=bioproject_sra_all&from_uid=1116726
https://ncbi.nlm.nih.gov/bioproject?Db=biosample&DbFrom=bioproject&Cmd=Link&LinkName=bioproject_biosample&LinkReadableName=BioSample&ordinalpos=1&IdsFromResult=1116726
UNITE database have more recent paper: https://doi.org/10.1093/nar/gkad1039

Experimental design

Figure 3A might have a contrast colors.
L206: Figure number is missing

Validity of the findings

L206: this statement requires explanation
L126, L253: what is known about the taxonomy of obtained strains? Are the ITS sequences of them available public?
The Discussion section is lacking the comparison with the previous strawberry microbiome studies.
The impact of pathogen to strawberry microbiome diversity and structure is interesting future direction.

Additional comments

L281: “fungal” → “Fungal”
L236: OTU or ASV?

Reviewer 2 ·

Basic reporting

- It was not found appropriate to give the obtained endophytic fungi at genus level. Although molecular identification was made using different primers, the species of the obtained fungi were not determined.
- It was not found appropriate to take samples from a single greenhouse. In fact, it was not understood that 30 plants were taken from three different varieties in just one greenhouse. It was also not appropriate.
- The use of three varieties in the variety reaction was not found appropriate.Variety reaction studies should be conducted with more varieties.
- There is no mention of the variety reaction in the methods subheading in the abstract. The writing format of the article also does not comply with the journal rules.
-It is unnecessary to give detailed information about organisms such as "Wood Saprotroph, Dung Saprotroph, Animal Pathogen-Plant Pathogen-Undefined Saprotroph" obtained in the study other than endophytes. It is irrelevant. Only endophyte species should have been focused on.
- The title "White Elves had stronger resistance to C. gloeosporioides and A. alternata" in the results is not appropriate. This is a result sentence, not a title.
- It is not stated where the fungi "Colletotrichum gloeosporioides and Alternaria alternata" in the manuscript were obtained. In addition, these 2 leaf pathogens were selected when there are more destructive pathogens in strawberry (such as Botrytis sp., Fusarium spp.). It would have been more appropriate to select a leaf and a fruit or root pathogen. If the fungi "Colletotrichum gloeosporioides and Alternaria alternata" were obtained in this study, how were they isolated from the collected healthy plants?
- In the results and discussion sections, it is not appropriate to compare endophyte fungi with general taxon groups such as Basidiomycota and Ascomycota rather than with species names. It should have been discussed on a species basis.
-The study's similarity rate was found to be high (33%)
-

Experimental design

No comment

Validity of the findings

No comment

Additional comments

The manuscript was not deemed appropriate for publication.

·

Basic reporting

Topic:
Endophytic fungal community composition and function response to strawberry genotype and disease resistance (#110358)

Manuscript Number:

110358

Reviewer’s Report:
1- First of all, I would like to thank the scientists for the scientific study
2- The topic is suitable for the manuscript.
3- Methodology and Materials are appropriate for the manuscript.
4- English language is clear
5- Statistical analyses were done.
6- Writing of the cultivar name must be like: cv. White Elves or ‘White Elves’ and so… . Not direct writing.
7- Line 118: ripampicin (50 g/L). Is it true? It may be 50 mg/L. Please check it.
8- Line 187: (Fig. S1) did not show the results. Please check it and write (Fig. 1, or 2, or…..) and (Table S1, or S2, or …….)
9- Line 206: (Fig. AC-B). I think it is wrong writing. It can be (Fig. 2A-B). Please check it.
10- Line 214: (Fig.S2) did not show the results. Please check it and write (Fig. 1, or 2, or…..) and (Table S1, or S2, or …….)
11- Figure 4C is absent in the text. Please check it.
12- Line 272: Talaromyces wushanicus must be italized.
13- All references have used in the text, but (Koljalg et al. 2013a) in the reference list but absent in the text. Please check it.

Experimental design

The manus. is clearly defined and meaningful.

Validity of the findings

Coclusion are well stated.

---

## Round 0.2 · Minor Revisions

Your manuscript needs minor revisions

·

Basic reporting

I would like to thank the authors for the improving the manuscript, but some concerns remain to be addressed.

Experimental design

L301-311: the composition of microbial fungal community is still discussing a phylum level, which is limited approach. Authors should distinguish the endophytic species against others.

Validity of the findings

The stated aim to “characterize the involvement of the plant genotype and it related disease tolerance on the fungal microbiomes of strawberry plant” isn’t reached in the Discussion section. Authors limit the analysis to relative abundances and major taxonomic groups, ignoring the microbiome structure, that could be divided to core and satellite parts. The results of multidimensional analysis are hardly described (L223-228). The impact of the genotype factor isn’t evaluated (see https://doi.org/10.3389/fmicb.2024.1453699 for method section).

Additional comments

Minor text issues:
L77: change “microbial biopreparations” to more suitable term
L81: “tollerance.” – correct typo

·

Basic reporting

Topic:
Endophytic fungal community composition and function response to strawberry genotype and disease resistance (#110358)

Manuscript Number:

110358

Reviewer’s Report:

The scientists did all correction
So, the manuscript can be accept.

Aysun Çavuşoğlu

Experimental design

OK

Validity of the findings

Topic:
Endophytic fungal community composition and function response to strawberry genotype and disease resistance (#110358)

Manuscript Number:

110358

Reviewer’s Report:

The scientists did all correction
So, the manuscript can be accept.

Aysun Çavuşoğlu

Additional comments

No

---

## Round 0.3 · Minor Revisions

After you make the changes below, your manuscript will be accepted.

L305: I suggest to check the literature about possible role of Malassezia fungus (which can be contamination from human skin).
L332-335: please rephrase to avoid repetiotion, is this hypothesis supported by previous studies of strawberry microbiome?
Additional comments
L121: Tween 20 is a trade mark

·

Basic reporting

pass

Experimental design

pass

Validity of the findings

L305: I suggest to check the literature about possible role of Malassezia fungus (which can be contamination from human skin).
L332-335: please rephrase to avoid repetiotion, is this hypothesis supported by previous studies of strawberry microbiome?

Additional comments

L121: Tween 20 is a trade mark

---

## Round 0.4 · accepted · Accept

Your manuscript is now eligible for acceptance.